# OpenReview forum: "Proximalized Preference Optimization for Diverse Feedback Types: A Decomposed Perspective on DPO"
_NeurIPS.cc/2025/Conference — NeurIPS 2025 poster_

### Official Review · Reviewer_iesQ · 2025-06-30

**Clarity:** 3
**Significance:** 4
**Originality:** 3
**Rating:** 5
**Confidence:** 4

**Summary:**

This paper addresses the limitation that Direct Preference Optimization (DPO) oversimplifies the regularization term. To highlight this issue, the authors present a series of theoretical results. In response, they propose Proximalized Preference Optimization (PRO), a variant of DPO designed to resolve likelihood underdetermination and provide a unified framework for aligning language models across diverse types of preference feedback. Theoretical analysis and empirical experiments are conducted to demonstrate the effectiveness of PRO.

**Questions:**

1. The corollary establishes a bound involving a constant $C$, but it would be helpful if the authors could further clarify the valid range or typical values of $C$, and how it depends on properties of the policy or data distribution.
2. In Theorem 4.2, $\alpha_0$ is given as a lower bound on the regularization coefficient $\alpha$. However, if $\alpha$ is required to be too large (i.e., $\alpha_0$ is large), then the regularization term may dominate the PRO loss, potentially hindering learning from the comparison data. Could the authors provide more discussion on how $\alpha_0$ is determined
3. If I understand correctly, the existence of an interior solution in Theorems 4.2 and 4.3 seems to address the degeneracy problem of the policy learned via the DPO loss, as highlighted in the [IPO](https://arxiv.org/abs/2310.12036) paper. Could the authors clarify whether this connection is intentional, and if so, provide experimental results or discussion to support this claim?
4. Theorem 3.1 shows the equivalence between the $\mathcal L_{eDPO}$ and the standard DPO formulation. While the formal result is clear, it would be helpful if the authors could provide additional intuition behind this equivalence.

**Ethical Concerns:**

["NO or VERY MINOR ethics concerns only"]

**Final Justification:**

The authors have addressed all my concerns through new experimental results and a clearer intuition for the proposed formulation. I maintain my score of acceptance.

**Limitations:**

The discussion of limitations is included in Appendix C.

**Quality:**

4

**Strengths And Weaknesses:**

Strengths:
1. The paper clearly identifies a key limitation of sample-based DPO methods, namely, the oversimplification of the regularizer term, which leads to likelihood underdetermination.
2. To address this issue, the authors propose a principled variant of DPO, called Proximalized Preference Optimization (PRO). They support their method with theoretical analysis and empirical results demonstrating its effectiveness.

Weaknesses:
1. Some of the theoretical proofs lack rigor. For example, the proof of Theorem 3.2 could be more formally structured by applying the KKT conditions.
2. Several theorems are presented without sufficient explanation or intuition, making them difficult to interpret. (see Questions section for specifics)

---

> ### Author Rebuttal · Authors · 2025-07-31
>
> Thanks for your insightful comments and helpful questions!
>
> >**W1: Some of the theoretical proofs lack rigor. For example, the proof of Theorem 3.2 could be more formally structured by applying the KKT conditions.**
>
> **A1**: We would like to clarify why the KKT conditions are not directly applied, and how our derivation still captures their underlying principles.
>
> The classical KKT conditions are designed for optimization problems with equality and **non-strict inequality** (i.e., $\\geq$ or $\\leq$) constraints. However, the problem in Theorem 3.2 is defined over the **open simplex** $\\Delta=\\{\\pi\\mid\\pi(y)>0,\\forall y\\in\\mathcal{Y}\\text{ and }\\sum\_{y\\in\\mathcal{Y}}=1\\}$, since the logarithm in the loss requires all $\\pi(y)$ strictly positive. This makes the KKT conditions not directly applicable.
>
> Briefly, the KKT conditions consist of:
> - Primal feasibility: The optimum satisfies all original constraints.
> - Stationary & complementary slackness: At the optimum, the gradient of the objective is a linear combination of the gradients of **active** constraints (those satisfied with equality), signifying that any descent direction will violate some constraint.
> - Dual feasibility: Lagrange multipliers for inequality constraints are non-negative.
>
> In our case, provided that the optimal solution exists (as assumed in Theorem 3.2):
> - The optimum necessarily lies in the interior of $\\Delta$ (i.e., the two conditions following Line 677).
> - This in turn means that sufficiently small moves in any direction remain the strict inequality constraints satisfied, and thus these constraints are, in effect, **inactive**. Then, we only need to consider the equality constraint; the stationarity condition reduces to that the gradient of the objective is colinear to the gradient of this equality constraint—precisely as stated in equation (16).
> - The variable $\\lambda$ in (16) serves as the Lagrange multiplier associated with this equality constraint and, in line with the KKT framework, is unconstrained.
>
> Therefore, each element above corresponds exactly, in a one-to-one fashion, to a component in the KKT framework. Based on this derivation, we further establish Theorem 3.2 by analyzing $\\lambda$, showing that it must be zero.
>
> We appreciate the reviewer’s attention to the rigor of our proofs and are glad to provide further clarification if needed.
>
> >**Q1: The corollary establishes a bound involving a constant $C$, but it would be helpful if the authors could further clarify the valid range or typical values of $C$, and how it depends on properties of the policy or data distribution.**
>
> **A2**: The constant $C$ depends on the values of $\\pi^\*(y)$ for $y$ where $\\hat{\\mu}(y)=0$ or $\\hat{s}(y)=0$. However, as established in Theorem 3.2, $\\pi^\*$ is determined by an equation system involving the non-linear sigmoid and logarithm functions, which makes a closed-form analysis of $C$ intractable.
>
> To gain more intuition on what influences $C$, recall that eDPO includes a regularizer to encourage $\\pi$ to remain close with $\\pi\_\\text{ref}$, alongside an optimizer that adjusts response probabilities upward or downward depending on the preference signal. According to (2), the preference signal is amplified by a factor of $\\hat{\\mu}(y)/\\mu(y)$ at optimum. If the dataset contains much more preferred responses with large amplification factors than dispreferred responses, the increases in probabilities for these preferred responses is likely to outweigh the decreases for dispreferred ones. Since probabilities must sum to one, this results in lower probabilities for unobserved responses—thus, $C$ tends to be smaller. In general, the typical scale of $C$ is inversely related to the total amplified factors of both preferred and dispreferred responses present in the data.
>
> Moreover, It is important to emphasize that **the main purpose of Corollary 3.3 is to show that eDPO enforces an ordering on the probability changes among preferred, unobserved, and dispreferred responses**. This means that the probability of unobserved response cannot be increased or decreased arbitrarily, as it is bounded by the changes to preferred and dispreferred responses. In other words, the corollary ensures that the underdetermination issue is resolved, regardless of the specific value of $C$.
>
> >**Q2: If the required $\\alpha$ is too large (i.e., $\\alpha_0$ is large), then the regularization term may dominate the PRO loss, potentially hindering learning from the comparison data. Could the authors provide more discussion on how $\\alpha_0$ is determined?**
>
> **A3:** We would like to clarify that $\\alpha$ and $\\beta$ jointly affect the strength of regularization relative to the learning signal. To elaborate, let us rewrite the PRO loss as
> $$
> \\frac{\\hat{\\mathcal{L}}\_\\text{PRO}}{\\beta}=-\\mathbb{E}\_{y\\sim\\hat{\\mu}}\\big[\\hat{s}(y)\\cdot\\log\\pi\_\\theta(y)\\big] + \\mathbb{E}\_{y\_1,y\_2\\dot{\\sim}\\mu}[f\_{\\alpha,\\beta}(\\delta)], \\text{ where } f\_{\\alpha,\\beta}(\\delta)=\\frac{\\alpha}{2\\beta}D\_\\text{KL}\\big(\\mathcal{B}(0.5)||\\mathcal{\\sigma(\\beta\\delta)}\\big) \\text{ and } \\delta = \\log\\frac{\\pi(y\_1)}{\\pi\_\\text{ref}(y\_1)}-\\log\\frac{\\pi(y\_2)}{\\pi\_\\text{ref}(y\_2)}.
> $$
> Here, $\\alpha$ and $\\beta$ only appear in $f\_{\\alpha,\\beta}$. To better understand their roles, note that
> $$\\nabla\_\\delta f\_{\\alpha,\\beta}(\\delta)=\\frac{\\alpha}{2}\\bigg[\\frac{1}{2}\\sigma(\\beta\\delta)-\\frac{1}{2}\\sigma(-\\beta\\delta)\\bigg]=\\frac{\\alpha}{2}\\bigg[\\sigma(\\beta\\delta)-\\frac{1}{2}\\bigg].
> $$
> This reveals that $\\alpha$ determines the **maximum gradient magnitude** of the regularizer, while $\\beta$ controls **how rapidly the gradient grows** as $\\delta$ moves away from zero. Therefore, even if a large $\\alpha$ is required, the overall regularization effect can be modulated by adjusting $\\beta$.
>
> We have empirically validated this in our response A1 to Reviewer 5FRF, where additional experiments are conducted by increasing $\\alpha$ beyond the one already producing strong performance. The results show that, even for $3\\times$ or $9\\times$ values of $\\alpha$, reducing $\\beta$ appropriately maintains competitive performance.
>
> These results also indicate that, once $\\alpha$ exceeds the required threshold, its further increase has minimal effect, as we can always find a suitable $\\beta$ to retain strong performance. Thus, in practice, it suffices to choose any $\\alpha$ above the certain threshold, without the need for precise tuning. Considering this, we present a constructive method to determine a sufficient $\\alpha\_0$ in our response A3 to Reviewer CDcH. With this analytical $\\alpha\_0$, hyperparameter tuning primarily focused on $\\beta$—similar to the standard DPO. For details, we kindly refer the reviewer to that response due to space constraints.
>
> >**Q3: The existence of an interior solution in Theorems 4.2 and 4.3 seems to address the degeneracy problem of the policy learned via the DPO loss, as highlighted in the IPO paper. Could the authors clarify whether this connection is intentional, and if so, provide experimental results or discussion to support this claim?**
>
> **A4**: Thanks for the insightful observation! While our initial derivation was not specifically intended to address the degeneracy issue highlighted in the IPO paper, Theorems 4.2 and 4.3 do guarantee that our proposed method resolves this problem.
>
> To empirically demonstrate, we conducted the same experiment as presented in the IPO paper: a bandit environment with three actions and the dataset $\\mathcal{D}=\\{y\_a\\succ y\_b, y\_b\\succ y\_c, y\_a\\succ y\_c\\}$. The policy to be optimized is parameterized as $\\pi\_\\theta(y\_i)=\\text{softmax}(\\theta)\_i$, with the reference policy being uniform. We trained the policy for 18,000 steps using Adam with learning rate $0.01$ and batch size 9. All methods converged before training completed; the summarized results are as follows:
> - PRO (we only present the result of $\\alpha=5$ due to the space constraints):
> |$\\beta\\qquad\\qquad$|$1.0\\qquad\\qquad$|$0.3\\qquad\\quad$|$0.1\\qquad\\quad$|
> |---|---|---|---|
> |$\\pi(y\_a)$|$0.4736$|$0.7701$|$0.9849$|
> |$\\pi(y\_b)$|$0.3173$|$0.1827$|$0.0149$|
> |$\\pi(y\_c)$|$0.2091$|$0.0472$|$0.0002$|
> - IPO:
> |$\\beta\\qquad\\qquad$|$1.0\\qquad\\qquad$|$0.3\\qquad\\quad$|$0.1\\qquad\\quad$|
> |---|---|---|---|
> |$\\pi(y\_a)$|$0.4469$|$0.6950$|$0.9647$|
> |$\\pi(y\_b)$|$0.3224$|$0.2295$|$0.0341$|
> |$\\pi(y\_c)$|$0.2307$|$0.0755$|$0.0012$|
> - DPO:
> |$\\beta\\qquad\\qquad$|$1.0\\qquad\\qquad$|$0.3\\qquad\\quad$|$0.1\\qquad\\quad$|
> |---|---|---|---|
> |$\\pi(y\_a)$|$1.0000$|$1.0000$|$1.0000$|
> |$\\pi(y\_b)$|$0.0000$|$0.0000$|$0.0000$|
> |$\\pi(y\_c)$|$0.0000$|$0.0000$|$0.0000$|
>
> As shown above, DPO degenerates to assigning all probability mass to response $a$, regardless of the $\\beta$ value. In contrast, both IPO and our PRO method maintain meaningful, non-trivial distributions over the actions.
>
> We will incorporate the above discussion into the revision.
>
> >**Q4: It would be helpful if the authors could provide additional intuition behind the equivalence in Theorem 3.1.**
>
> **A5**: Theorem 3.1 essentially utilizes an interesting property of the log-sigmoid function. Concretely,
>
> \begin{align}
> \\nabla\_\\delta\\big[a\\log\\sigma(\\delta)+(1-a)\\log\\sigma(-\\delta)\\big]&=a\\sigma(-\\delta)-(1-a)\\sigma(\\delta)\\\\
> &=a-\\sigma(\\delta)=\\nabla\_\\delta\\big[a\\delta+\\log\\sigma(-\\delta)\\big]\\\\
> &=a-1+\\sigma(-\\delta)=\\nabla\_\\delta\\big[(a-1)\\delta+\\log\\sigma(\\delta)\\big].
> \end{align}
>
> In other words, the convex combination of log-sigmoid gradients can be decoupled into two parts: one that depends on the data-relevant signal $a$; another that is independent of $a$, and thus can be re-arranged to serve as a regularizer.
>
> We will incorporate the above explanation in the revision to better convey the intuition underlying our proposed method.

---

> > ### Comment · Reviewer_iesQ · 2025-08-04
> >
> > Thank you for the thorough response. All my concerns are addressed, so I will keep my current score for acceptance.

---

> > > ### Author Response · Authors · 2025-08-04
> > >
> > > We are pleased to hear that all the raised concerns have been addressed. Thanks again for your valuable comments!

---

### Official Review · Reviewer_o7Vr · 2025-07-01

**Clarity:** 3
**Significance:** 3
**Originality:** 2
**Rating:** 4
**Confidence:** 4

**Summary:**

This work identifies the issue of likelihood underdetermination in direct alignment methods for LLMs, which causes decreased absolute likelihoods of example responses and undesired outputs. It revisits and reformulates the loss in direct preference optimization (DPO), revealing the oversimplification of a regularizer as the underlying cause. Based on this, the authors introduce PRO, which resolves the underdetermination issue through an efficient approximation of the complete regularizer and demonstrates superior performance in handling various feedback types.

**Questions:**

See the section above

**Ethical Concerns:**

["NO or VERY MINOR ethics concerns only"]

**Final Justification:**

This work addresses the issue of likelihood underdetermination within direct alignment methods for Large Language Models (LLMs). This problem leads to a decrease in the absolute likelihood of example responses and results in undesired outputs. By revisiting and reformulating the loss function of Direct Preference Optimization (DPO), the authors identify an oversimplified regularizer as the root cause. Building on this insight, they introduce Preference Ranking Optimization (PRO), which resolves the underdetermination issue by efficiently approximating the complete regularizer, demonstrating superior performance across various feedback types. All my concerns are addressed.

**Limitations:**

See the section above

**Paper Formatting Concerns:**

Appendices are missing.

**Quality:**

3

**Strengths And Weaknesses:**

Strengthes:
1. The paper provides a novel decomposed reformulation of the Direct Preference Optimization (DPO) loss function. This reformulation broadens the applicability of DPO to a wider range of feedback types, and offers new insights into the underlying cause of likelihood underdetermination.
2. The authors identify that the standard DPO implementation oversimplifies a regularizer, leading to likelihood underdetermination. They demonstrate that reinstating the complete regularizer effectively resolves this issue.

Weaknesses and Questions:
1. Appendices seem to be missing.
2.  It is surprising to see the bad performance of DPO. It would be helpful The paper does not provide detailed information on how hyperparameters were chosen. More transparency in the hyperparameter tuning process would enhance the reproducibility and credibility of the results.

---

> ### Author Rebuttal · Authors · 2025-07-31
>
> Thank you for your review and feedback!
>
> We would like to clarify that all appendices are included in the **supplementary material**, in line with NeurIPS guidelines, which state that extensive appendices can be provided either in the supplementary material or the main submission. For your convenience, the appendices cover the following sections:
> - A &nbsp; Related Work . . . . . . . . . . . . . . . . . . . . . . . . . . . . . . . . . . . . . . . . 15
> - B &nbsp; Proof of Theorems . . . . . . . . . . . . . . . . . . . . . . . . . . . . . . . . . . . 16
> - C &nbsp; Comparison of PRO Loss and RLHF Objective . . . . . . . . . . . . 25
> - D &nbsp; Implementation Details of PRO Loss . . . . . . . . . . . . . . . . . . . . 28
> - E &nbsp; Performance Degeneration of KTO . . . . . . . . . . . . . . . . . . . . . 29
> - F &nbsp; The Role of $\\alpha$ and $\\beta$ in PRO Regularizer . . . . . . . . . . . . . . . . . . 29
> - G &nbsp; Additional Experimental Setup . . . . . . . . . . . . . . . . . . . . . . . . . 29
> - H &nbsp; Additional Experimental Results . . . . . . . . . . . . . . . . . . . . . . . . 31
>
> We understand how important complete appendices are for a thorough evaluation, and apologize for any confusion regarding their location. We would be grateful if you could further assess our submission in light of the availability of appendices.
>
> Below, we respond to the remaining concerns raised in the review.
>
> >**Q1: It is surprising to see the bad performance of DPO.**
>
> **A1**: We agree that DPO's underperformance on the Anthropic-HH dataset is noteworthy. We believe this is due to the following differences between our setup and the original DPO paper:
> - **Evaluation Criteria**: Unlike the DPO paper, which assesses helpfulness alone, our evaluation incorporates helpfulness, harmlessness, and conciseness. As noted in [1], the Anthropic-HH dataset is annotated for both helpfulness and harmlessness (with HH being short for them). Moreover, LLM judges are known to favor overly verbose responses; several benchmarks (e.g., AlpacaEval 2) and previous works (e.g., KTO) have explicitly included conciseness as an evaluation dimension. These metrics therefore more closely match the intent of the dataset as well as real-world alignment objectives.
> - **Sequence Length**: Our experiments utilize more modern context windows (prompt: 1024, sequence: 2048) compared to the original DPO setting (256/512 respectively). As reward hacking often manifests as generating overly verbose outputs (please see the right panel of Figure 1), increasing the context window could further exacerbate this phenomenon. Given that modern LLMs typically operate with even longer contexts, DPO’s vulnerability to such issue is likely even greater.
>
> Furthermore, the underwhelming results of DPO we observe are consistent with earlier findings in the KTO paper [2], where Figure 2 shows that SFT alone outperforms SFT+DPO across multiple Pythia model sizes.
>
> We will revise the manuscript accordingly to reflect these points.
>
> [1] Training a helpful and harmless assistant with reinforcement learning from human feedback, 2022.
> [2] Model alignment as prospect theoretic optimization, 2024.
>
> >**Q2: The paper does not provide detailed information on how hyperparameters were chosen. More transparency in the hyperparameter tuning process would enhance the reproducibility and credibility of the results.**
>
> **A2**: We appreciate the reviewer’s concern regarding transparency in hyperparameter tuning. To clarify:
> - **General hyperparameters**: These are specified in Appendix G (Line 954) and align with the official KTO repository (commit ID: 306ed27). We did not perform additional tuning beyond these canonical settings.
> - **Method-specific hyperparameters ($\\alpha$, $\\beta$)**:
>   - **$\\beta$ selection**: We conduct a hyperparameter sweep over $\\beta$, as described in Lines 324-326. Figure 2 summarizes the results across $\\beta\\in\\{0.1, 0.03, 0.01, 0.003, 0.001\\}$ on the Anthropic-HH dataset. For the UltraFeedback dataset, Table 1 reports the best performance across these $\\beta$ values for each method, as described in Lines 332-335. For imbalanced binary feedback (from the Anthropic-HH dataset), we adopt the optimal $\\beta$ identified in Figure 2, as mentioned in Line 345-346 and 358-359. For scalar feedback, we similarly perform a sweep over the same range of $\\beta$; we will add this clarification in the revision.
>   - **$\\alpha$ selection**: Details of $\\alpha$ settings are provided in Appendix D (Lines 907-909, 917-918), and a reference is also provided in Line 301 of the main paper. Additionally, based on the comment from Reviewer 5FRF, we have added further experiments during the rebuttal phase to examine the performance across a range of $\\alpha$ (see our response A1 to Reviewer 5FRF).
>
> We hope these clarifications sufficiently address the reviewer's concerns. If there are any additional points or specific aspects the reviewer would like us to elaborate on, we would be glad to provide additional clarification.

---

> > ### Author Response · Authors · 2025-08-06
> >
> > Dear Reviewer o7Vr,
> >
> > As the discussion phase is nearing its end, we would like to kindly follow up and confirm whether our responses have addressed your concerns. If there's anything unclear or if you'd like us to elaborate further, we're eager to do so.
> >
> > Thanks once again for your time and insights on our work!

---

### Official Review · Reviewer_5FRF · 2025-07-02

**Clarity:** 2
**Significance:** 3
**Originality:** 3
**Rating:** 5
**Confidence:** 4

**Summary:**

This paper presents Proximalized Preference Optimization (PRO), a new method for aligning large language models (LLMs) with human feedback that addresses key limitations of the widely used Direct Preference Optimization (DPO) method. The authors first identify a critical issue in DPO they term "likelihood underdetermination," where the model's likelihood for both preferred and dispreferred responses decreases during training. This can lead to reward-hacking effects, causing the model to generate outputs that deviate from expected patterns. The core contribution is a theoretical reformulation of the DPO loss function, decomposing it into two distinct parts: an optimizer term that reorganizes feedback into a pointwise signal, naturally extending its applicability beyond pairwise comparisons to other formats like binary and scalar feedback; and a regularizer term that is independent of preference labels. Through this new perspective, the paper argues that likelihood underdetermination in standard DPO arises from an oversimplification of this regularizer. The proposed PRO method addresses this by incorporating a more complete, yet computationally efficient, approximation of the regularizer. This is achieved through a novel "hyper response" mechanism, which groups unobserved responses to make the calculation tractable. PRO offers a unified framework for aligning LLMs with diverse feedback types (pairwise, binary, and scalar) while simultaneously resolving the underdetermination issue.Extensive experiments demonstrate that PRO successfully mitigates reward hacking and achieves performance that is comparable to or better than specialized methods like DPO, KTO, and NCA across various feedback scenarios, including challenging cases with extremely imbalanced data.

**Questions:**

1. I think this paper does not define $\hat{\mu}$ clearly. I think $\hat{\mu}$ is the responses with feedbacks, and $\mu$ is all responses. Is that correct? I think the authors should state this more clearly around Line 164. Otherwise, what is "full regularizer"?
2. Why Corollary 3.3 solves the underdetermination issue? I think the underdetermination is the important weight diminishing. But the difference of probability ratios of chosen and rejected pairs could still be large under Corollary 3.3.
3. I'm not sure I understand "unobserved"correctly. I think it means responses without feedbacks. So Sec 4.1 is basically you treat unobserved responses evenly in the regularizer's expectation. I think you could make this easier to understand for readers.
4. I think you use hyper response because sampling  ${y_1,y_2\sim{\mu}}$ is intractable. But is this Hyper Response method only be able to use in PRO? I think in the  population-based formulation of DPO (Line 144), you can also use Hyper Response, is that correct?

I give the score "Borderline reject". But I am willing to raise my score if my questions are resolved.

**Ethical Concerns:**

["NO or VERY MINOR ethics concerns only"]

**Final Justification:**

The rebuttal clarify my concern on the hyperparameter selection, underdetermination, importance weight diminishing, and hyper responses. I think this is a very good paper. Writing needs some improvement to make it clearer to follow.

**Limitations:**

yes

**Quality:**

3

**Strengths And Weaknesses:**

### Strengths
- This paper first proves that the population-based DPO's gradient is equivalent to the eDPO's gradient. The reformulation decomposes the loss into an optimizer and a regularizer, and only the optimizer relies on the preference feedbacks. This decomposition therefore provides greater flexibility in developing sample-based loss since the preference feedbacks are usually limited.
- A major contribution is the unification of alignment across pairwise, binary, and scalar feedback within a single framework. While methods like KTO and NCA were developed for specific feedback types , PRO's design, derived from its reformulated optimizer, naturally accommodates this diversity.


### Weaknesses
- Some sentences are not easy to understand. See questions.
- PRO introduces a new hyperparameter, $\alpha$, which balances the optimizer and regularizer. While the experiments on imbalanced data show this to be a powerful and necessary lever, it also adds to the complexity of hyperparameter tuning compared to standard DPO. From Table 2, the impact of $\alpha$ is significant. How to select the best $\alpha$? The paper could be strengthened by including a more systematic sensitivity analysis for $\alpha$ across different settings to provide practitioners with better guidance.
- The "hyper response" is an elegant approximation, but it is still an approximation. It prevents the model from differentiating the probabilities of individual responses within the hyper set. The paper argues this is not problematic because the total probability mass is constrained. However, it is possible that this could mask certain failure modes where the model learns to assign disproportionate probability to a specific "bad" but unobserved response.
- In Table 2, other baseline methods are missing.

---

> ### Author Rebuttal · Authors · 2025-07-31
>
> Thanks for the constructive feedback and helpful questions!
>
> >**W1: How to select the best $\\alpha$? The paper could be strengthened by including a more systematic sensitivity analysis for $\\alpha$ across different settings to provide practitioners with better guidance.**
>
> **A1**: We conducted additional experiments to analyze the sensitivity of PRO to $\\alpha$, with the results summarized as follows.
> - Win rate (%) under pairwise feedback:
> |$\\alpha~ \\backslash~\\beta\\quad$|$1e^{-2}\\quad$|$3e^{-3}\\quad$|$1e^{-3}\\quad$|$3e^{-4}\\quad$|$1e^{-4}$|
> |---|---|---|---|---|---|
> |$2.5$|$51.47\\quad$|$\\mathbf{53.21}\\quad$|$49.71\\quad$|
> |$7.5$||$51.87$|$\\mathbf{52.97}$|$49.03$|
> |$22.5$|||$51.21$|$\\mathbf{53.65}$|$50.13$|
> - Win rate (%) under 1%-desired binary feedback:
> |$\\alpha~ \\backslash~\\beta\\quad$|$1e^{-2}\\quad$|$3e^{-3}\\quad$|$1e^{-3}\\quad$|$3e^{-4}\\quad$|$1e^{-4}$|
> |---|---|---|---|---|---|
> |$17.5$|$48.91$|$\mathbf{57.21}$|$52.13$|
> |$52.5$||$49.83$|$\mathbf{56.51}$|$51.49$|
> |$157.5$|||$48.31$|$\mathbf{56.33}$|$53.09$|
>
> For reference, in our paper, $\\alpha=2.5$ for pairwise feedback and $\alpha=17.5$ for 1%-desired binary feedback are sufficiently large to yield strong performance.
>
> In the new experiments, we further increased $\\alpha$ by factors of 3 and 9 to examine its impact. The results show that, once $\\alpha$ surpasses the required threshold, there always exists a suitable $\\beta$ such that the competitive performance is maintained. In other words, the performance is insensitive to further increase in $\\alpha$ beyond this point. This raises the opportunity to simplify the hyperparameter tuning. Due to the space constraints, we refer the reviewer to our response A2 to Reviewer CDcH, where we provide an explanation for the above phenomenon, and discuss a practical hyperparameter tuning strategy that brings its overhead on par with DPO.
>
> >**W2: The "hyper response" is an elegant approximation, but it is still an approximation. It prevents the model from differentiating the probabilities of individual responses within the hyper set. The paper argues this is not problematic because the total probability mass is constrained. However, it is possible that this could mask certain failure modes where the model learns to assign disproportionate probability to a specific "bad" but unobserved response.**
>
> **A2**: We would like to clarify that enumerating every unlabeled response in alignment is fundamentally computationally intensive, and this limitation is not unique to PRO. In fact, existing approaches such as DPO, KTO, NCA, and IPO only include labeled responses in their losses, leaving all other unobserved responses unconstrained. As a result, due to the absence of training signals, the probabilities of unobserved responses may drift arbitrarily during optimization, which potentially leads to specific "bad" and unobserved response being assigned with disproportionately high probability. PRO makes this limitation transparent, rather than introducing a new one.
>
> Moreover, even if additional computation for unlabeled responses were feasible, existing methods do not specify how to include them in the loss. By contrast, PRO allows for straightforward exclusion of these responses from the hyper-response set, and the regularizer can directly optimize their probability mass. This provides a flexible way to balance computational cost and regularization granularity across the response space.
>
> >**W3: In Table 2, other baseline methods are missing.**
>
> **A3**: Among the baselines, only KTO supports binary feedback. Its results, based on using the optimal $\\beta$ identified in Figure 2, are reported in Lines 345-347 and 358-359 (since the primary focus of Table 2 is to investigate how $\\alpha$ affects PRO’s performance, we did not include them there).
>
> In the 1%-desired setting, KTO performs poorly and appears to suffer from reward hacking: the aligned model generates numerous duplicated and meaningless tokens (Line 347). We thus increased $\\beta$ in an attempt to bring the optimized model closer to the reference model, but the results remained unsatisfactory (Line 350). For this reason, we did not report the win rates in the submission. We provide the corresponding results below and will include them in the revision:
> |$\\beta$|$0.03\\qquad$|$0.1\\qquad$|$0.3\\qquad$|$1\\qquad~~$|$3$|
> |---|---|---|---|---|---|
> |Win rate (\%)|$22.46$|$27.29$|$18.13$|$15.21$|$16.89$|
>
> If the reviewer has more suggestions for adapting other baselines to binary feedback, we are happy to include them in the revision.
>
> >**Q1: I think this paper does not define $\\hat{\\mu}$ clearly. I think $\\hat{\\mu}$ is the responses with feedbacks, and $\\mu$ is all responses. Is that correct? I think the authors should state this more clearly around Line 164. Otherwise, what is "full regularizer"?**
>
> **A4**: We agree that this distinction is critical and will explicitly clarify it around Line 164. Specifically:
> - $\\hat{\\mu}$ refers to the empirical distribution over responses that are labeled with feedbacks (i.e., empirical distribution of the preference dataset, as described in Line 167).
> - $\\mu$ refers to the distribution over all possible responses (typically, it can be the underlying distribution from which responses are sampled for preference annotation, as stated in Line 89).
>
> >**Q2: Why Corollary 3.3 solves the underdetermination issue? I think the underdetermination is the important weight diminishing. But the difference of probability ratios of chosen and rejected pairs could still be large under Corollary 3.3.**
>
> **A5**:  Let us clarify the terminology and the role of Corollary 3.3 detailedly：
> - **Underdetermination** means that adding a constant to both log-probabilities of the preferred and dispreferred responses ($a\rightarrow a+c, b\rightarrow b+c$) does not alter the value of DPO loss:
> $$
> \\log\\sigma\\big((a+c)-(b+c)-\\log\\pi_{ref}(y_w)+\\log\\pi_{ref}(y_l)\\big) = \\log\\sigma\\big(a-b-\\log\\pi_{ref}(y_w)+\\log\\pi_{ref}(y_l)\\big),
> $$
> As a result, when $c$ is negative, the probability mass can "leak" from the labeled responses to the unobserved responses, causing the "reward hacking" phenomenon observed in prior works.
> - **Importance weight diminishing** provides a more specific explanation of how, in practice, both preferred and dispreferred log-probabilities decrease in DPO. Specifically, once the log-probability gap is large enough, there is no incentive to update the model, even if the log-probabilities of preferred responses have dropped (which highly likely happens due to catastrophic forgetting).
>
> In other words, underdetermination creates the possibility for reward hacking, while importance weight diminishing makes it actually occur during DPO training.
>
> Corollary 3.3 addresses the underdetermination by enforcing an order among the log-probability changes after optimization. Specifically, it ensures that the log-probability change for any unobserved response must lie between those for preferred and dispreferred responses. Unlike before, the simultaneous probability decreases for preferred and dispreferred responses would also require a reduction in the probability of unlabeled responses, which is impossible due to the constraint of fixed total probability.
>
> Lastly, we note that the log-probability difference between paired responses becoming unboundedly large is a separate issue of DPO, often referred to as degeneracy. We appreciate that the comment (and Reviewer iesQ's Q3) for bringing our attention to the fact that PRO also addresses this issue. For further details, please refer to our response A4 to Reviewer iesQ.
>
> >**Q3: I'm not sure I understand "unobserved" correctly. I think it means responses without feedbacks. So Sec 4.1 is basically you treat unobserved responses evenly in the regularizer's expectation. I think you could make this easier to understand for readers.**
>
> **A6**: "Unobserved" indeed refers to the responses without feedbacks, i.e., those not represented in the preference dataset.
>
> To clarify, treating unobserved responses evenly in the regularizer's expectation would still require enumerating every such response, which is computationally infeasible. In Section 4.1, we aggregate all unobserved responses into a single hyper response, treating it as a unified group. This allows us to reduce the expectation to just $N+1$ terms: $N$ for the labeled responses and one for the hyper response. Computation of the regularizer then requires evaluating $\\mu$, $\\pi$, and $\\pi\_\\text{ref}$ on the hyper response. To enable it, we define $p(\\mathcal{H})=1-\\sum\_{y\\notin\\mathcal{H}}p(y)$ for any distribution $p$. This avoids the need for explicit enumeration and makes the computation tractable.
>
> We will revise accordingly to make these points easier to understand.
>
> >**Q4: I think you use hyper response because sampling $y\_1,y\_2\\in\\mu$ is intractable. But is this Hyper Response method only able to use in PRO? I think in the population-based formulation of DPO (Line 144), you can also use Hyper Response, is that correct?**
>
> **A7**: We appreciate this insightful question. Hyper response is indeed designed to address the intractability of sampling $y\_1,y\_2\\in\\mu$. According to Theorem 4.1, hyper response can only be constructed from unlabeled/unobserved responses. In contrast, the population-based DPO loss assumes access to preference feedback for every pair, which means that hyper response cannot be directly applied to it.
>
> However, as stated in Theorem 4.3, under the specific choices of $\\mu$ and $\\alpha$, PRO recovers a similar form with DPO, except that it is based on an augmented empirical preference and **involves the hyper-response mechanism**. Actually, it can be treated as an enhanced variant of sample-based DPO, where pseudo preference $p(y\_1\succ y\_2)=1/2$ is introduced for $y\_1,y\_2\\in\\mathcal{Y}\_\\mathcal{H}\\setminus\\text{supp}(\\hat{\\mu})$.

---

> > ### Author Response · Authors · 2025-08-06
> >
> > Dear Reviewer 5FRF,
> >
> > As the discussion phase is nearing its end, we would like to kindly follow up and confirm whether our responses have addressed your concerns. If there's anything unclear or if you'd like us to elaborate further, we're eager to do so.
> >
> > Thanks once again for your time and insights on our work!

---

> ### Comment · Reviewer_5FRF · 2025-08-08
>
> Thanks for your reply. It resolves my concerns. My suggestion includes clarify "underdetermination" and "Importance weight diminishing" in the final draft. I raise my score to "Accept". One additional question: can you give me an example of "hyper response"? How is the hyper response constructed?

---

> ### Author Response · Authors · 2025-08-08
>
> Thank you sincerely for your positive feedback and for raising the score. We appreciate your helpful suggestions and will make sure to clarify "underdetermination" and "importance weight diminishing" in the final draft as you advised.
>
> **Regarding the question on "hyper response":**
> The hyper response is a *conceptual aggregate* representing unlabeled responses. In practice, we do not need to explicitly enumerate its elements for approximating the regularizer. We illustrate this below with a simple example.
>
> Assume we have only a pair of labeled responses, denoted by $y\_w$ and $y\_l$. Given that the hyper response consists of all unlabeled responses, the overall response space is $\\{y\_w, y\_l, \\mathcal{H}\\}$. In the regularizer, the expectation over all pairs can be explicitly written as (omitting constant terms):
> \\begin{align}
> &\\frac{\\alpha}{2}\\mathbb{E}\_{y\_1,y\_2\\dot\\sim\\mu}\\bigg[D\_\\text{KL}\\bigg(\\mathcal{B}\\bigg(\\frac{1}{2} \\bigg) \~\\Bigg|\\Bigg|\~ \\mathcal{B}\\Big(\\sigma\\big(r\_\\theta(y\_1) - r\_\\theta(y\_2)\\big)\\Big) \\bigg]  \\\\
> &=  \\alpha \\mu(y\_w)\\mu(y\_l)\\bigg[D\_\\text{KL}\\bigg(\\mathcal{B}\\bigg(\\frac{1}{2} \\bigg) \~\\Bigg|\\Bigg|\~ \\mathcal{B}\\Big(\\sigma\\big(r\_\\theta(y\_w) - r\_\\theta(y\_l)\\big)\\Big) \\bigg]  \\\\
> & \\quad +  \\alpha \\mu(y\_w)\\mu(\\mathcal{H})\\bigg[D\_\\text{KL}\\bigg(\\mathcal{B}\\bigg(\\frac{1}{2} \\bigg) \~\\Bigg|\\Bigg|\~ \\mathcal{B}\\Big(\\sigma\\big(r\_\\theta(y\_w) - r\_\\theta(\\mathcal{H})\\big)\\Big) \\bigg]  \\\\
> & \\quad +  \\alpha \\mu(y\_l)\\mu(\\mathcal{H})\\bigg[D\_\\text{KL}\\bigg(\\mathcal{B}\\bigg(\\frac{1}{2} \\bigg) \~\\Bigg|\\Bigg|\~ \\mathcal{B}\\Big(\\sigma\\big(r\_\\theta(y\_l) - r\_\\theta(\\mathcal{H})\\big)\\Big) \\bigg],  \\\\
> \\end{align}
> where
> \\begin{gather}
> \\mu(\\mathcal{H}) = 1 - \\mu(y\_w) - \\mu(y\_l),   \\\\
> r\_\\theta(\\mathcal{H}) = \\beta\\log\\frac{\\pi\_\\theta(\\mathcal{H})}{\\pi\_\\text{ref}(\\mathcal{H})} = \\beta\\log\\frac{1-\\pi\_\\theta(y\_w) - \\pi\_\\theta(y\_l)}{1-\\pi\_\\text{ref}(y\_w) - \\pi\_\\text{ref}(y\_l)}.
> \\end{gather}
> This enables us to succinctly represent and efficiently compute the regularizer, without instantiating every possible unlabeled response.
>
> We will further clarify and illustrate the function of the hyper response in the revision. Thank you again for your insightful question.

---

> > ### Comment · Reviewer_5FRF · 2025-08-08
> >
> > Thanks for your response. I notice that you imply this in Line 224-226, but you should make it clearer for the first time readers.

---

> ### Author Response · Authors · 2025-08-08
>
> Thank you for this constructive suggestion! We plan to add a figure to visually illustrate how the modified response space and the approximated expectation relate to the original ones. We believe this will make our explanation more accessible.

---

### Official Review · Reviewer_CDcH · 2025-07-03

**Clarity:** 3
**Significance:** 3
**Originality:** 3
**Rating:** 5
**Confidence:** 3

**Summary:**

This paper proposes proximalized preference Optimization (PRO) by reformulating the standard DPO. The reformulation incorporates a complete regularizer to fix likelihood underdetermination and robustly align large language models across pairwise, binary, and scalar feedback. To make the optimization scalable, PRO uses a hyper-response approximation to efficiently maintain absolute likelihoods. It outperforms existing methods by preventing reward hacking and ensuring stable, high-quality model alignment.

**Questions:**

1.	Can the $\alpha_0$ in Thm 4.2 be determined in practice?
2.	How to compute the $f(H)$ in line 223? This also corresponds to $r(H)$.
3.	How is the hyperresponse $H$ constructed?

**Ethical Concerns:**

["NO or VERY MINOR ethics concerns only"]

**Final Justification:**

The authors have addressed my concerns. I raised my score from 4 to 5.

**Limitations:**

Yes.

**Paper Formatting Concerns:**

No.

**Quality:**

3

**Strengths And Weaknesses:**

**Strengths**:

1.	The paper studies an equivalent form of DPO and answers the question of why standard DPO has the problem of likelihood underdetermination, which stems from the oversimplified regularizer.

2.	The new loss derived can now handle multiple types of feedback, including pairwise, binary, and scalar feedback. Therefore, it is a unified framework.

3.	RPO is robust to extreme data imbalance.

4.	The training of RPO is still contrastive, and therefore inherits the advantage of DPO that no reward model training is needed.


**Weaknesses**:

1.	The idea of hyperresponse is interesting and saves compute. However, I am concerned that this also forces rare or novel responses to share one probability mass, which can suppress diversity.

2.	RPO now introduces an additional hyperparameter $\alpha$, which also needs to be tuned jointly with $\beta$. Therefore, the tuning might become harder in practice. From the imbalanced experiment, the performance of RPO seems to be affected by $\alpha$.

---

> ### Author Rebuttal · Authors · 2025-07-31
>
> Thanks for your thoughtful comments and appreciating the idea of hyperresponse!
>
> >**W1: The idea of hyperresponse is interesting and saves compute. However, I am concerned that this also forces rare or novel responses to share one probability mass, which can suppress diversity.**
>
> **A1**: Hyperresponse aggregates a group of unlabeled responses, which prevents us from distinguishing their individual probabilities. However, as established in Theorem 4.1, this aggregation does not alter the total probability mass assigned to this group in the optimal solution; only the allocation within the group is not explicitly managed. This means that while rare or novel responses may sometimes receive low probabilities, the effect arises from limited intra-group control, not from increased competition among these responses.
>
> We would like to further clarify two important points:
> - The inability to control the distribution over unlabeled responses is a challenge shared by all existing alignment methods, not one caused by hyperresponse. Prior approaches such as DPO, KTO, NCA, and IPO optimize the probabilities only for labeled responses, leaving the rest under-specified. Hyperresponse simply makes this explicit and transparent, instead of introducing a new issue.
>
> - Accounting for the individual probabilities of unlabeled responses would require explicitly computing them, which is intrinsically computationally intensive. Moreover, existing methods—even if computational resources allow—typically do not incorporate these probabilities into the loss function. In contrast, PRO moves a forward step by offering a flexible manner to tradeoff the computational cost and the granularity in the probability allocation over the response space. Specifically, if certain unlabeled responses are deemed important (e.g., we want to preserve the model’s capacity for generating them), they can be excluded from the aggregation. In this way, the regularizer would be computed over a finer-grained response space, preserving more detailed likelihood knowledge from the reference model.
>
> We appreciate this valuable comment and will incorporate the above discussion in the revision.
>
> >**W2: PRO now introduces an additional hyperparameter $\\alpha$, which also needs to be tuned jointly with $\\beta$. Therefore, the tuning might become harder in practice. From the imbalanced experiment, the performance of PRO seems to be affected by $\\alpha$.**
>
> **A2**: We would like to clarify that, in practice, tuning $\\alpha$ and $\\beta$ is considerably simpler than it might appear. Specifically, it is not necessary to jointly tune $\\alpha$ and $\\beta$: any value of $\\alpha$ above a certain threshold suffices; for each such $\\alpha$, there exist corresponding $\\beta$ values that allow our method to consistently attain strong performance. We first provide a detailed explanation for this assertion, followed by supporting empirical evidence.
>
> By rewriting the PRO loss as
> $$
> \\frac{\\hat{\\mathcal{L}}\_\\text{PRO}}{\\beta}=-\\mathbb{E}\_{y\\sim\\hat{\\mu}}\\big[\\hat{s}(y)\\cdot\\log\\pi\_\\theta(y)\\big] + \\mathbb{E}\_{y\_1,y\_2\\dot{\\sim}\\mu}[f\_{\\alpha,\\beta}(\\delta)], \\text{ where } f\_{\\alpha,\\beta}(\\delta)=\\frac{\\alpha}{2\\beta}D\_\\text{KL}\\big(\\mathcal{B}(0.5)||\\mathcal{\\sigma(\\beta\\delta)}\\big) \\text{ and } \\delta = \\log\\frac{\\pi(y\_1)}{\\pi\_\\text{ref}(y\_1)}-\\log\\frac{\\pi(y\_2)}{\\pi\_\\text{ref}(y\_2)},
> $$
> the hyperparameters $\\alpha$ and $\\beta$ are only involved in the function $f\_{\\alpha,\\beta}$. This allows us to analyze their impacts on the loss by simply examining $f\_{\\alpha,\\beta}$. Noticing that $\\nabla\_\\delta f\_{\\alpha,\\beta}(\\delta)=\\frac{\\alpha}{2}\\big[\\frac{1}{2}\\sigma(\\beta\\delta)-\\frac{1}{2}\\sigma(-\\beta\\delta)\\big]=\\frac{\\alpha}{2}\\big[\\sigma(\\beta\\delta)-\\frac{1}{2}\\big]$, we see that $\\alpha$ determines the **maximum gradient magnitude** while $\\beta$ governs **how rapidly the gradient grows** as $\\delta$ departs 0. If $\\alpha$ is too small, the gradient of the overall loss can be dominated by the optimizer, causing unpreferred responses reduced towards zero probability and thus compromising our theoretical guarantees. Theorem 4.2 established there exists some $\\alpha_0$: for any $\\alpha>\\alpha_0$, the regularizer maintains its effectiveness and prevents probabilities from vanishing.
>
> Importantly, a value of $\\alpha$ being sufficiently large implies that the magnitude of the regularizer gradient should rarely hit its saturation regime during optimization. Now, let $\\alpha_1$ be a sufficient large value (i.e., $\\alpha\_1>\\alpha\_0$), and $\\beta\_1$ denote the tuned value accordingly. As $\\alpha$ and $\\beta$ provides adequate flexibility to shape the curve of $\\nabla\_\\delta f\_{\\alpha, \\beta}$, for any $\\alpha_2>\\alpha_1$, we can always select a $\\beta_2$ such that the curve of $\\nabla\_\\delta f\_{\\alpha\_2, \\beta\_2}$ shows a very similar shape with $\\nabla\_\\delta f\_{\\alpha\_1, \\beta\_1}$ prior to its saturation region. For instance, in Figure 3 (right column) of Appendix F, the gradient curves for $(\\alpha=1, \\beta=3)$ and $(\\alpha=3, \\beta=1)$ largely overlap in the region of $\\delta\\in(-0.5, 0.5)$. In this way, the optimization procedure would progress similarly under these two settings.
>
> Empirical evidence (please see our response A1 to Reviewer 5FRF for details) also confirms that given sufficiently large $\\alpha$ there always exists suitable $\\beta$ to make PRO achieve similar performances. In other words, once $\\alpha$ exceeds a required threshold, it has marginally effect on the performance.
>
> Therefore, the tuning process involves first determining a sufficiently large $\\alpha$, after which $\\beta$ can be tuned independently. Theorem 4.3 provides a valid threshold of $\\alpha$ in the case of pairwise feedback, and we will elaborate another one for general form of feedback in A3. With that, we only need to tune $\\beta$, and the tuning overhead becomes similar to that of DPO.
>
> >**Q1: Can the $\\alpha\_0$ in Thm 4.2 be determined in practice?**
>
> **A3**: Yes—$\\alpha\_0$ can *constructively* determined in practice.
>
> As shown in the proof of Theorem 4.2 (Line 773), to prevent PRO loss from decreasing indefinitely as the solution approaches **a specified boundary point** $\\pi_\\infty$ of the feasible region, it suffices to select
> $$
> \\alpha>\\frac{4\\hat{\\mu}(y^0)\\cdot(-\\hat{s}(y^0))}{\\mu(y^0)\\mu(\\mathcal{Y}^+)}, \\quad \\forall y^0\\in\\mathcal{Y}^0:\\hat{s}(y^0)<0,
> $$
> where
> $$
> \\mathcal{Y}^0 = \\big\\{y\\mid\\pi\_\\infty(y)=0, y \\in\\mathcal{Y}\_\\mathcal{H}\\big\\}\\quad\\text{and}\\quad\\mathcal{Y}^+ =\\big\\{y\\mid \\pi\_\\infty(y)>0, y\\in\\mathcal{Y}\_\\mathcal{H}\\big\\}.
> $$
> Once $\\alpha$ satisfies this condition for **all boundary points**, the loss cannot decrease continuously on the whole boundary, thus ensuring the existence of an optimal solution within the feasible region.
>
> This can be achieved by further strengthening the above condition to make it $\\pi\_\\infty$-independent:
> - Instead of enforcing the inequality only for $y^0\\in\\mathcal{Y}^0: \\hat{s}(y^0)<0$, we can require it for all $y^0\\in\\mathcal{Y}\_\\mathcal{H}:\\hat{s}(y^0)<0$.
> - Since $\\mu(\\mathcal{Y}^+)=\\sum\_{y\\in\\mathcal{Y}^+}\\mu(y)\\geq \\min\_{y\\in\\mathcal{Y}^+}\\mu(y) >\\min\_{y\\in\\mathcal{Y}\_\\mathcal{H}}\\mu(y)$, we can safely use this lower bound for further simplification.
>
> Putting these together, a sufficient and easily computable choice is
> $$
> \\alpha\_0=\\max\_{y\\in\\mathcal{Y}\_\\mathcal{H}:\\hat{s}(y)<0}\\bigg[\\frac{4\\hat{\\mu}(y)\\cdot(-\\hat{s}(y))}{\\mu(y)\\cdot\\min\_{y'\\in\\mathcal{Y}\_\\mathcal{H}}\\mu(y')}\\bigg].
> $$
>
> *PS*: As discussed in A2, the value of $\\alpha$ does not perceptibly affect the performance, once it exceeds the required threshold. Therefore, there is no need to pursue the minimal $\\alpha\_0$.
>
> >**Q2: How to compute the $f(\\mathcal{H})$ in Line 223? This also corresponds to $r(\\mathcal{H})$.**
>
> **A4**: The general function $f$ is used to illustrate how expectation can be efficiently computed under the hyperresponse approximation (as described in Line 223). In the case of PRO loss, $f$ is instantiated as a function involving the terms $\\mu(\\mathcal{H})$ and $r(\\mathcal{H})=\\beta\\log\\frac{\\pi(\\mathcal{H})}{\\pi\_\\text{ref}(\\mathcal{H})}$ with respect to $\\mathcal{H}$. By the definition (6), the relevant quantities are calculated as
> $$
> \\mu(\\mathcal{H})=1-\\sum\_{y\\notin\\mathcal{H}}\\mu(y),\\quad\\pi(\\mathcal{H})=1-\\sum\_{y\\notin\\mathcal{H}}\\pi(y),\\quad\\pi\_\\text{ref}(\\mathcal{H})=1-\\sum\_{y\\notin\\mathcal{H}}\\pi\_\\text{ref}(y).
> $$
> Since $\\mathcal{H}$ is typically constructed (in Lines 215-219) to include all unlabeled responses, the complement of $\\mathcal{H}$ corresponds exactly to the labeled responses. As their probabilities are necessarily computed for alignment, computing the above terms incurs negligible additional cost.
>
> >**Q3: How is the hyperresponse $\\mathcal{H}$ constructed?**
>
> **A5**: Theorems 4.1-4.3 require that $\\mathcal{H}$ consists exclusively of unlabeled responses. By default, we construct it by including all such unlabeled responses (in Lines 215-219) , and this construction is consistently used in our reported experiments.
>
> In initial experiments on the HH dataset, we also explored the effect of excluding 2 or 4 unlabeled and $\\pi\_\\text{ref}$-generated responses from $\\mathcal{H}$. The results show negligible changes in win rate (less than 0.5%).
>
> However, in scenarios where certain unlabeled responses generated from the reference model are of particular interest, excluding them from $\\mathcal{H}$ can allow for more targeted regularization, and help maintain the model’s ability to generate these responses.

---

> > ### Author Response · Authors · 2025-08-06
> >
> > Dear Reviewer CDcH,
> >
> > As the discussion phase is nearing its end, we would like to kindly follow up and confirm whether our responses have addressed your concerns. If there's anything unclear or if you'd like us to elaborate further, we're eager to do so.
> >
> > Thanks once again for your time and insights on our work!

---

> ### Comment · Reviewer_CDcH · 2025-08-07
>
> Thanks for the authors' responses. I still have one more question left regarding the sufficient choice of
>
> $\alpha_0 = \max_{y \in Y_H: \hat{s}(y) < 0} \left[ \frac{4\hat{\mu}(y) \cdot (-\hat{s}(y))}{\mu(y) \cdot \min_{y' \in Y_H} \mu(y')} \right]$. Since the $\max$ and $\min$ here are within response space $Y_H$, can these be easily done?

---

> ### Author Response · Authors · 2025-08-07
>
> Thank you for raising this insightful follow-up question.
>
> To clarify, $\mathcal{Y}\_\\mathcal{H}=\\{\\mathcal{H}\\}\\cup\\{y\\mid y\\notin\\mathcal{H}\\}$ denotes the modified response space after introducing the hyper response $\\mathcal{H}$. Typically, $\\mathcal{H}$ acts as a single aggregated response that contains all unobserved responses, and $\\{y\\mid y\\notin\\mathcal{H}\\}$ directly correspond to the labeled responses. Therefore, the cardinality satisfies $|\\mathcal{Y}\_\\mathcal{H}|=N+1$, where $N$ is the number of labeled responses, and the extra $1$ accounts for the hyper response.
>
> Since $N$ is usually small in practice (e.g., just a few per prompt), the $\\max$ and $\\min$ over $\\mathcal{Y}\_\\mathcal{H}$ can be efficiently computed by direct enumeration.
>
> We appreciate the reviewer’s attention to implementation details and will clarify this point explicitly in the revision to preclude any ambiguity.

---

> > ### Comment · Reviewer_CDcH · 2025-08-08
> >
> > Thanks for the further explanation. I don't have any concern now.

---

> ### Author Response · Authors · 2025-08-08
>
> Thank you for your follow-up. It’s great to hear that all concerns have been addressed. We appreciate your time and valuable feedback!

---

### Comment · Area_Chair_sReN · 2025-08-06

Dear Reviewers,

Thank you for your initial reviews. I’d like to remind everyone to actively engage in the author–reviewer discussion (and thank you if you’ve already done so!).

- If authors have resolved your (rebuttal) questions, do tell them so.

- If authors have not resolved your (rebuttal) questions, do tell them so too.

As per NeurIPS review policy this year, please make sure to submit the “Mandatory Acknowledgement” **only after you have read the rebuttal and participated in the discussion**.

Thank you for your efforts,

AC

---

### Note · Authors · 2025-08-13

We sincerely thank the reviewers for thoughtful feedback and the area chair for facilitating a constructive discussion.

We are pleased to note that **Reviewers CDcH, 5FRF, and iesQ confirmed our responses fully addressed their concerns**. For Reviewer o7Vr, we did not receive a follow-up during the discussion.

**Reviewer o7Vr’s main concern** appeared to be missing appendices. In response, we clarified all appendices were included in the supplementary material per NeurIPS guidelines. On **the remaining concerns**—DPO underperformance and hyperparameter tuning details—we noted:
- Our evaluations use more comprehensive criteria and modern sequence lengths compared to the DPO paper; similar DPO underperformance was also observed in prior work (see A1).
- Tuning procedures are detailed in the main paper and appendices, with references in A2.

We believe these clarify that the appendix concern was a misunderstanding, and the other points were about experimental clarity but not core limitations.

Beyond individual points, **reviewers widely agreed on the paper's contributions and strengths**:
- A novel and principled reformulation of DPO that links likelihood underdetermination to an oversimplified implicit regularizer (CDcH, 5FRF, o7Vr, iesQ).
- An interesting and elegant hyper-response mechanism that approximates the complete regularizer and resolves underdetermination (CDcH, 5FRF).
- A unified alignment framework covering pairwise, binary, and scalar feedback (CDcH, 5FRF, o7Vr, iesQ).
- Supporting theory (iesQ) and extensive experiments showing resistance to reward hacking (CDcH, 5FRF), robustness to extreme data imbalance (CDcH), and strong performance across feedback types (5FRF, o7Vr).

Reviewer iesQ also insightfully noted that our method resolves degeneracy—another issue of DPO reported in prior work; we substantiate this with empirical results (A4 to iesQ Q3).

Finally, to **position our work in the field of LLM alignment/post-training**, we highlight:
- We recast the widely discussed likelihood drop after alignment as likelihood underdetermination, pinpoint its root cause via an equivalent DPO reformulation, and propose a practical resolution.
- By unifying loss across feedback types, our approach enables alignment to scale with richer signal sources.
- The reformulation itself suggests an intriguing bridge to RLHF, promising milder regularization, improved model diversity, and integration with better-calibrated preference models (see Appendix C).

---

### Decision · Program_Chairs · 2025-09-17

**Decision:**

Accept (poster)

**Comment:**

This paper proposes Proximalized Preference Optimization (PRO), a variant of DPO designed to resolve likelihood underdetermination and provide a unified framework for aligning language models across diverse types of preference feedback. Reviewers appreciated the clear decomposed formulation, theoretical grounding, and promising empirical results. Given the overall positive sentiment, I recommend acceptance.